# Effect of restricting bedtime mobile phone use on sleep, arousal, mood, and working memory: A randomized pilot trial

**Jing-wen He[1]☯, Zhi-hao Tu[2]☯, Lei Xiao[1], Tong Su[1], Yun-xiang Tang[1]***

**1** Department of Medical Psychology, the Second Military Medical University, Shanghai, China,
**2** Department of Nautical Psychology, the Second Military Medical University, Shanghai, China

☯ These authors contributed equally to this work.
* plostang@sina.com

## Abstract

### Background

This study aimed to assess the effects of restricting mobile phone use before bedtime on sleep, pre-sleep arousal, mood, and working memory.

### Methods

Thirty-eight participants were randomized to either an intervention group (n = 19), where members were instructed to avoid using their mobile phone 30 minutes before bedtime, or a control group (n = 19), where the participants were given no such instructions. Sleep habit, sleep quality, pre-sleep arousal and mood were measured using the sleep diary, the Pittsburgh sleep quality index, the Pre-sleep Arousal Scale and the Positive and Negative Affect Schedule respectively. Working memory was tested by using the 0-,1-,2-back task (n-back task).

### Results

Restricting mobile phone use before bedtime for four weeks was effective in reducing sleep latency, increasing sleep duration, improving sleep quality, reducing pre-sleep arousal, and improving positive affect and working memory.

### Conclusions

Restricting mobile phone use close to bedtime reduced sleep latency and pre-sleep arousal and increased sleep duration and working memory. This simple change to moderate usage was recommended to individuals with sleep disturbances.

**Data Availability Statement:** All relevant data are within the manuscript and its Supporting Information files.

**Funding:** This work was supported by the Mental Health Application Research of PLA (12XLZ109) and the National Natural Science Foundation of China (81372122) to YT. The funders had no role in study design, data collection and analysis, decision to publish, or preparation of the manuscript.

**Competing interests:** The authors have declared that no competing interests exist.

# Introduction

Mobile phones have become an essential tool in our daily life in the past decade. The number of mobile phone users in China has increased from 0.39 billion in 2012 to 0.72 billion in 2017, accounting for 72.2% in 2012 and 96.3% of the netizens in 2017. Students are the largest group of Chinese Internet users [1]. Mobile phone use prior to bedtime or even after lights-out is a common habit among many young adults. However, this unhealthy habit may lead to delayed bedtime, sleep loss, irregular sleep-wake patterns, poor sleep quality, and increased tiredness during the day [2–4]. In addition, media content or games might induce pre-sleep hyper-arousal [5].

Sleep plays an important role in mood cognitive function. Sleep loss, sleep restriction, and sleep disorders might have a negative effect on mood and cognitive functions. Previous studies found that sleep disorders were associated with mood deficits [6], and sleep loss led to anger, confusion, anxiety, and depression [7–8]. In addition, sleep loss might impair working memory [9] and vigilant attention [10]. Furthermore, one meta-analysis found that individuals with insomnia exhibit performance impairments in several cognitive functions, including working memory and executive functioning [11].

Many cross-sectional studies have investigated the relationship between mobile phone use and sleep [12–15]. Nevertheless, regarding the effects of restricting mobile phone use around bedtime on sleep, the number of existing randomized-controlled trials is limited. Furthermore, these studies report contradictory results. To our knowledge, only Harris et al investigated the effect of restricting the use of electronic media after 22:00 in high school athletes, and they did not find any improvement in sleep habits, athletic performance, cognitive performance, or mood after four weeks of this intervention [16]. Meanwhile, another study assessed the impact of restricting phone use for one hour before bedtime and found that this restriction could advance lights-out time and increase total sleep-time [17].

Thus, in our study, we aimed to investigate the effects of restricting mobile phone use before bedtime in college students who have a habit of using a mobile phone before sleep and thus have poor sleep quality. We hypothesized that the restriction of mobile phone use will help reduce pre-sleep arousal, enable early sleep, and regulate a healthier sleep with good sleep quality. Additionally, we presumed that sleep will mediate mood and cognitive performances.

# Methods

## 2.1. Participants

Study participants were recruited from our university through social media from April 1st to April 8th 2019. The inclusion criteria were poor sleep quality (Pittsburgh sleep quality index [PSQI]>5) and a habit of using a mobile phone during bedtime (>30 mins). The exclusion criteria included the following: (1) self-reported medical conditions such as depression, schizophrenia, metabolic diseases, cardiovascular diseases, chronic or recurrent respiratory conditions, active cancer, or neurological disorders; (2) other self-reported sleep disorders such as obstructive sleep apnea-hypopnea syndrome, restless leg syndrome, or rapid eye movement sleep disorder; (3) use of medications, devices, or hypnotics to assist sleep; (4) excessive caffeine use (>500 mg/day) or excessive alcohol use (>4 cups/day); (5) shift workers or those with work requiring long-distance driving or operating heavy equipment; (6) self-reported total nap duration of >90 min/day or excessive daytime sleepiness; and (7) pregnancy, lactation, or perimenopausal state with irregular menses.

A total of 72 individuals were screened, and 32 were found to be ineligible. Out of these 32 individuals, 23 were ineligible because of the presence of other sleep disturbances, five

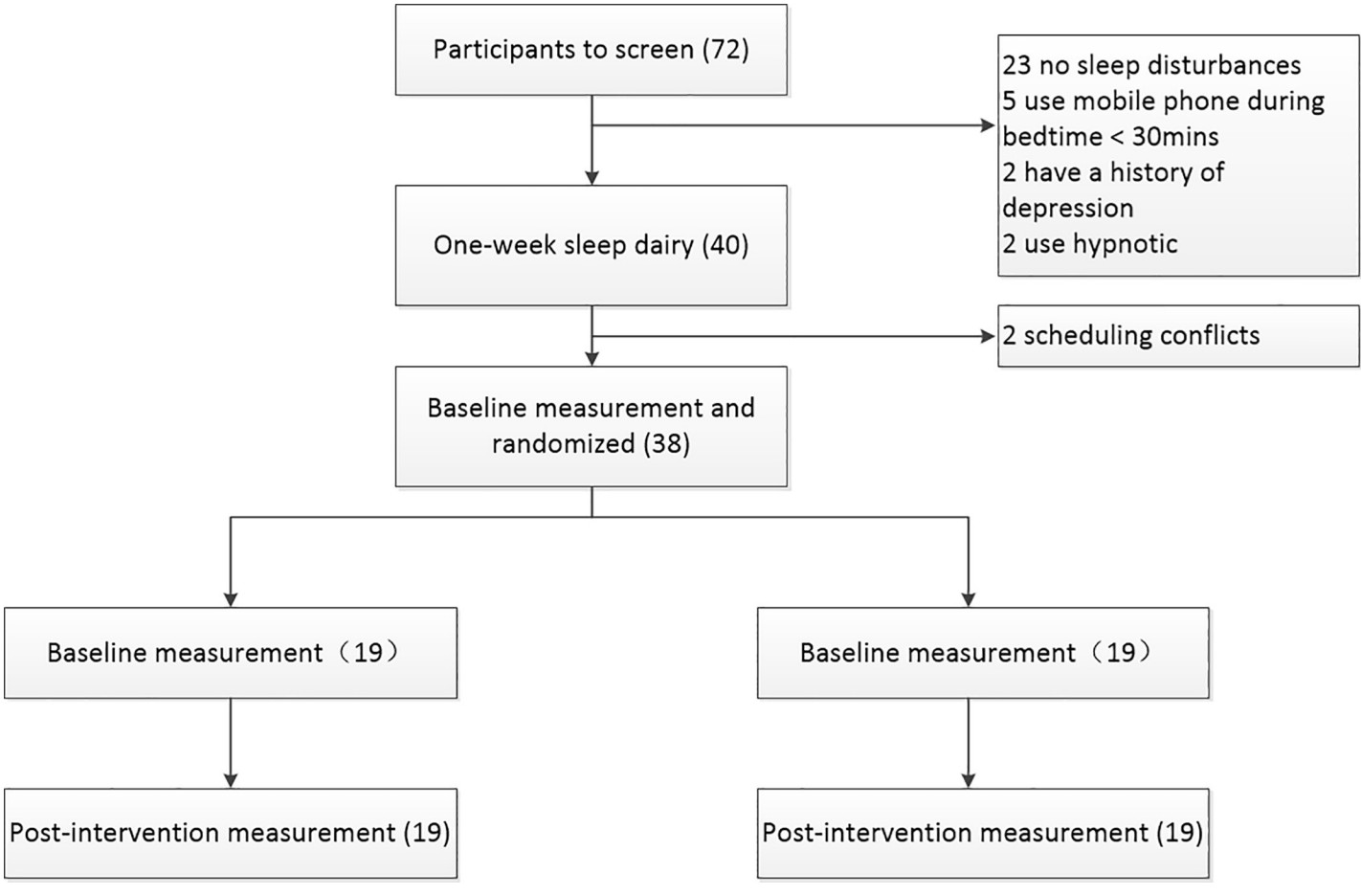

**Fig 1. Flow chart.**

volunteers were ineligible because of their mobile phone use habit during bedtime or for use for less than 30 minutes prior to time of sleep. Two participants were ineligible because of medication use to help with sleep and another two were ineligible because of a history of depression. After screening, two participants were dropped because they were being deployed in the South China sea for medical treatment. (Fig 1)

This study was approved by the Ethical Committee of the Second Military Medical University. A complete survey description was first presented to the participants. Written informed consent and oral approval were obtained before the baseline measurements were taken.

## 2.2. Intervention

All participants entered baseline data before intervention and data after a week's trial of intervention into an online sleep diary. Then, the participants of the intervention group were instructed to refrain from using their mobile phone for 30 minutes prior to their average bedtime. In the intervention group, 12 participants used Huawei, five used Apple, and two used Mi phones. For participants whose mobile phones were Huawei or Apple, we could set the screen time. Using the "manage screen time" function in the "settings," we set the mode as follows: "this is my child's phone" and then "bedtime" and then a password was added. Thus, during the set "bedtime," the screen would turn grey and no apps could be opened; the phone

goes off service after this set time. As for participants whose mobile phones were of other brands, they were instructed to turn them off at the appointed time every night. One researcher would send a message to remind them to turn off their mobile phones 10 minutes before their appointed time, and two researchers would call them at any time after turn-off time every night to ensure that everyone had turned off their mobile phones.

The participants in the control group did not receive any instructions regarding their sleep or mobile phone use.

## 2.3. Measures

Assessments were conducted at baseline and post-intervention.

**2.3.1. Sociodemographic variables.** The sociodemographic variables included age, sex, body mass index, marital status, exercise habits, smoking, and the frequency of drinking coffee, tea, or alcohol.

**2.3.2. Sleep quality measure.** The PSQI is a 19-item self-reported questionnaire that assesses sleep quality and disturbances over the past month. The total score comprises seven component scores: subjective sleep quality, sleep latency, sleep duration, habitual sleep efficiency, sleep disturbances, use of sleep medication, and daytime dysfunction. Scores range from 0 to 21 and a high score indicates poor sleep [18].

**2.3.3. Sleep diary.** Sleep diary data one week before and after the intervention were collected. An online sleep diary was sent to every individual by "Wechat" in the morning. The participants were instructed to complete the sleep diary as soon as they woke up. The sleep diary items included bedtime, sleep latency, wake time, rise time, sleep duration, mobile phone usage time per day, and mobile phone usage time prior to bedtime. Mobile phone usage time prior to bedtime was defined as phone use from 9:00 pm to the time of sleep. Sleep diaries are viewed as inexpensive and valid measures of sleep in healthy participants and are known to be highly correlated with data from actigraphy.

**2.3.4. Arousal measure.** Pre-sleep arousal was evaluated with the Pre-sleep Arousal Scale (PSAS) [19]. PSAS is a 16-item, self-reported questionnaire comprising both cognitive and somatic manifestations of arousal, with eight items in each subscale. Two subscale scores ranging from 8 to 40 were computed separately: Pre-sleep Arousal Scale Cognitive Arousal (PSAS-C) with sum items 1–8, and Pre-sleep Arousal Scale Somatic Arousal (PSAS-S) with sum items 8–16. When responding to the PSAS, subjects were asked to describe how intensely they generally experienced each component as they attempted to fall asleep in their own bedroom by selecting an appropriate rating of "1: do not at all," "2: slightly," "3: moderately," "4: a lot," or "5: extremely," during the past week.

**2.3.5. Mood measure.** The Positive and Negative Affect Schedule (PANAS) is a 20-item scale for measuring the positive and negative effects based on a 5-point Likert scale. Ten questions are related to the positive effects, including enthusiasm, pleasure, and engagement. A high score reflects a high degree of positive effect. Ten questions are related to the negative effects such as feeling distressed, upset, or guilty. A high score indicates a high degree of negative effect [20].

**2.3.6. Working memory.** In the n-back task, participants were shown a series of numbers presented one by one for a duration of 1000 milliseconds each and an intertrial interval of 700 milliseconds. This task involves three different memory load conditions: 0-, 1-, and 2-back. Participants respond by clicking the mouse button. In the 0-back condition, participants are instructed to click the right button every time a "five" is displayed and the left button when other numbers were presented. In the 1-back condition, they are instructed to click the right button when the number presented is identical to the previous one (e.g., a 5 followed by a 5)

and to click the left button if not identical (e.g., a 5 followed by a 4). In the 2-back condition, they are instructed to click the right button when the current number is identical to the one before the last (e.g., a 5 followed by a 1 and then another 5) and click the left mouse button when they are not identical. There were six blocks (two for each condition), each block consisting of 30 trials. Each block lasted for approximately two minutes, leading to a total testing time of 12 minutes. The practice consisted of 10 trials for all conditions, and once all the practice trials were correctly completed, participants were allowed to start the task. Outcome measures were the reaction time and accuracy.

## 2.4. Statistical analysis

Before performing statistical analyses, trials with reaction times (RTs) exceeding ±3 standard deviations (SD) from the mean were excluded as outliers and removed from the RT and accuracy analyses. Outliers comprised less than 1% of the trials for each group and condition.

No formal sample size calculations were made as this was a pilot study. Statistical analysis was performed with SPSS version 22.0. Descriptive statistics such as mean, range, SD, and percentages were calculated. We used the unpaired t-test to test group differences in baseline statistics. Between-subject analyses of variance, with one between-group factor (intervention vs control group) and one within-subjects/repeated measures factor (baseline vs follow-up), were conducted to assess the effect of the intervention on participants' scores on sleep, arousal, mood, and cognitive performance. The main effects on time and group, and the interaction effects (group × time) are reported. When the interaction effects were significant, a simple effect analysis was conducted.

## Results

The demographic information and characteristics of participants are reported in Table 1. No significant differences between the groups at baseline were observed.

### 3.1. Sleep

There was a significant interaction between groups and time regarding sleep latency, sleep duration, and PSQI scores (Table 2). Analysis of the simple effect test results for sleep latency showed a significant difference in the between-group post-test (p = 0.049) and a significant difference (p<0.01) over time in the intervention group (Fig 2A). Analysis of the simple effect test results for sleep duration showed a significant difference (p = 0.035) over time in the intervention group (Fig 2B). Analysis of the simple effect test results for PSQI scores showed a significant difference in the between-groups post-test (p<0.01) and a significant difference (p<0.01) over time in the intervention group (Fig 2C).

### 3.2. Pre-sleep hyperarousal

There was a significant interaction between groups and time regarding somatic and cognitive arousal (Table 2). Analysis of the simple main effect test results for somatic arousal showed a significant difference in the between-groups post-test (p = 0.008) and a significant difference (p<0.01) over time in the intervention group (Fig 3A). Analysis of the simple main effect test results for cognitive arousal showed a significant difference in the between-groups post-test (p = 0.001) and a significant difference (p<0.01) over time in the intervention group (Fig 2B).

**Table 1. Baseline demographics and clinical characteristics.**

|  | Intervention group N = 19 | Control group N = 19 | *P* |
|---|---|---|---|
| Sex |  |  | 0.728 |
| Male | 12(63.2%) | 14(73.7%) |  |
| Female | 7(36.8%) | 5(26.3%) |  |
| Age | 20.95±2.068 | 21.37±2.63 | 0.587 |
| Marital status |  |  | 0.486 |
| Single | 19(100%) | 17(89.5%) |  |
| Married | 0(0%) | 2(10.5%) |  |
| Coffee consumption |  |  | 0.115 |
| Never or seldom | 9(47.4%) | 12(63.2%) |  |
| Sometimes | 10(52.6%) | 5(26.3%) |  |
| Often or always | 0(0%) | 2(10.5%) |  |
| Tea consumption |  |  | 0.743 |
| Never or seldom | 7(36.8%) | 9(47.4%) |  |
| Sometimes | 12(63.2%) | 10(52.6%) |  |
| Often or always | 0 | 0 |  |
| Smoking status |  |  | 0.486 |
| Never or seldom | 19(100%) | 17(89.5%) |  |
| Sometimes | 0 | 0 |  |
| Often or always | 0 | 2(10.5%) |  |
| Alcohol consumption |  |  | 0.295 |
| Never or seldom | 15(78.9%) | 11(57.9%) |  |
| Sometimes | 4(21.1%) | 8(42.1%) |  |
| Often or always | 0 | 0 |  |
| Duration of mobile phone use in one day (hours) | 5.47±1.87 | 5.82±3.31 | 0.698 |
| Duration of mobile phone use during bedtime (hours) | 1.47±1.08 | 1.39±0.68 | 0.809 |

## 3.3. Mood

There was a significant interaction between groups and time for positive affect (Table 2). Analysis of the simple main effect test results showed a significant difference in the between-groups post-test (p = 0.01), and a significant difference (p = 0.001) over time in the intervention group (Fig 3C). There was an important effect on the group regarding the negative affect (Table 2). The results indicated that there was a significant difference in the between-group post-test (p = 0.06) and a significant difference (p = 0.037) over time in the intervention group (Fig 3D).

## 3.4. Working memory

No significant main effects of time and group, and interaction (group × time) was found in 0-back accuracy (Table 2 and Fig 4A). There was a significant main effect of time in 1-back accuracy (Table 2). The result indicated that there was a significant difference in the between-groups post-test (p = 0.046) and a significant difference (p = 0.04) over time in 1-back accuracy in the intervention group (Fig 4C). A significant main effect of time in 2-back accuracy was found (Table 2). The result indicated that there was a significant difference in between-group post-test (p = 0.036) and a significant difference (p = 0.03) over time in 2-back accuracy in the intervention group (Fig 4E).

A significant main effect of time in 0-back RT was detected (Table 2). Analysis of the simple main effect test results showed a significant difference in the between-groups post-test

**Table 2. Sleep, arousal, mood, and working memory by group over time.**

| | Intervention | | Control | | Time effect | Group effect | Time x group |
|---|---|---|---|---|---|---|---|
| | Pretest | Posttest | Pretest | Posttest | | | |
| Sleep latency | 30.98(14.22) | 18.38(11.85) | 29.32(10.13) | 26.54(12.78) | 16.34** | 0.85 | 6.67* |
| Sleep duration | 6.20(0.58) | 6.56(0.71) | 6.48(0.45) | 6.27(0.68) | 0.43 | 0.003 | 5.91* |
| PSQI | 10.05(2.51) | 5.63(1.64) | 9.63(1.64) | 8.58(1.57) | 31.47** | 7.71** | 11.92** |
| PSAS-S | 14.58(5.07) | 10.79(3.03) | 14.68(4.37) | 14.32(4.56) | 9.42** | 2.19 | 6.38* |
| PSAS-C | 23.32(7.23) | 16.37(6.07) | 24.21(4.72) | 22.05(3.55) | 23.52** | 4.55* | 6.51* |
| Positive affect | 26.53(7.13) | 31.53(6.39) | 26.58(5.48) | 26.00(6.12) | 4.82* | 2.36 | 7.68** |
| Negative affect | 18.26(5.80) | 14.84(4.68) | 21.58(5.19) | 20.74(7.50) | 3.632 | 8.84** | 1.33 |
| 0-back ACC | 0.99(0.016) | 0.98(0.017) | 0.98(0.016) | 0.98(0.031) | 1.635 | 1.41 | 0.093 |
| 1-back ACC | 0.94(0.055) | 0.97(0.018) | 0.94(0.070) | 0.95(0.038) | 5.017* | 0.94 | 0.604 |
| 2-back ACC | 0.93(0.072) | 0.96(0.042) | 0.91(0.066) | 0.93(0.046) | 7.174* | 2.85 | 0.257 |
| 0-back RT | 571.96(114.18) | 499.63(85.74) | 570.82(89.52) | 555.62(83.56) | 6.079* | 1.22 | 2.590 |
| 1-back RT | 651.23(118.84) | 581.69(98.36) | 634.34(93.08) | 634.64(82.76) | 4.810* | 0.41 | 4.892* |
| 2-back RT | 774.87(197.92) | 670.50(171.35) | 815.35(186.53) | 730.79(139.34) | 18.537** | 0.92 | 0.204 |

PSQI, the Pittsburgh Sleep Quality Index; PSAS-S, Pre-sleep Arousal Scale Somatic Arousal; PSAS-C, Pre-sleep Arousal Scale Cognitive Arousal; ACC, accuracy; RT, reaction time.

*p<0.05.

**p<0.01.

(p = 0.049) and a significant difference (p = 0.07) over time in 0-back RT in the intervention group (Fig 4B). For 1-back RT, there was a significant interaction between groups and with time (Table 2). Analysis of the simple main effect test results showed a significant difference (p = 0.004) over time in 1-back RT in the intervention group (Fig 4D). A significant main effect of time was found in 2-back RT (Table 2). Analysis of the simple main effect test results showed a significant difference (p = 0.002) over time in the intervention group and a significant difference (p = 0.01) over time in 2-back RT in the control group (Fig 4F).

## Discussion

In this study, the effects of restricting mobile phone use before bedtime on sleep, pre-sleep arousal, mood, and working memory were investigated. After a four-weeks of intervention, effective reduction of sleep latency, increased sleep duration, improved sleep quality, reduced pre-sleep arousal, reduced negative affect, and improved positive affect and working memory were achieved.

Sleep latency in the intervention group was reduced by around 12 minutes and sleep duration was increased by around 18 minutes. Sleep quality was improved significantly. This was consistent with the results of a previous study which found that adolescents who stopped using their phones 80 min earlier and turned their lights off 17 min earlier, slept 21 min longer after one week of phone usage restriction [17]. This implies that limiting the use of mobile phones before bedtime could effectively improve sleep. First, it could reduce the impact of light emitted by mobile phones on sleep. Second, reducing the arousal induced by mobile phone contents may also help sleep. This simple change to moderate use was recommended by media scholars.

Pre-sleep arousal was decreased significantly in the intervention group, suggesting that it could be reduced by restricting mobile phone usage around bedtime. Mobile phone content could induce pre-sleep arousal, which might lead to difficulty in falling asleep and poor sleep

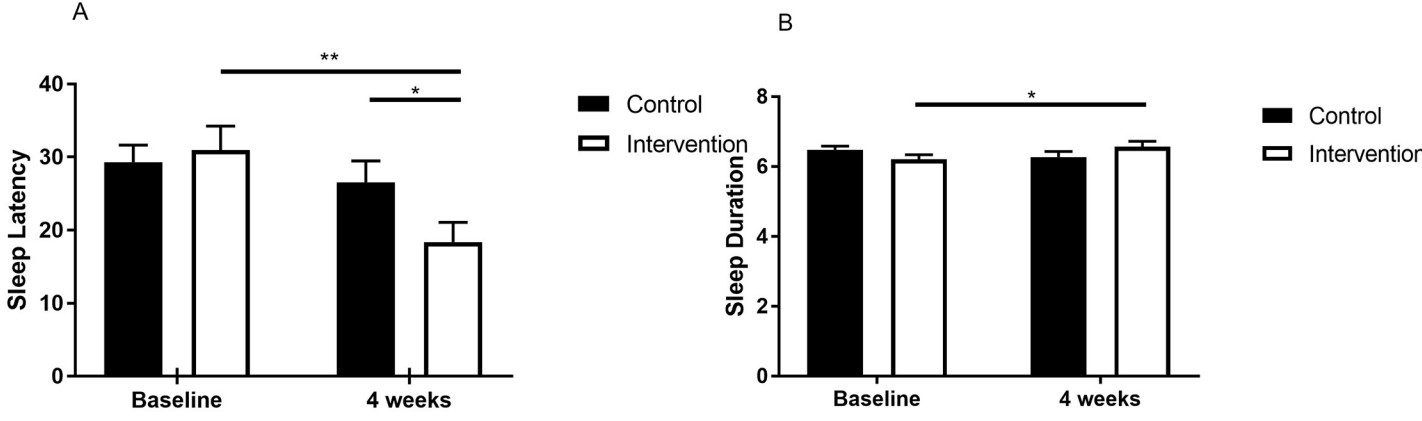

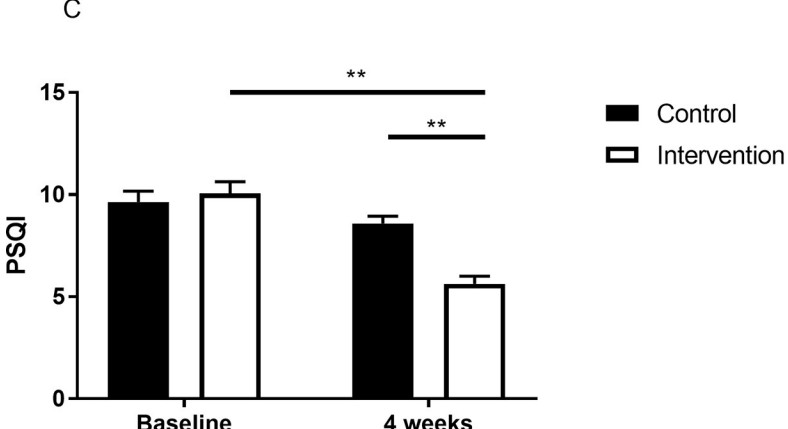

**Fig 2. The change in sleep after the intervention.** A, sleep latency at baseline and at the end of intervention. B, sleep duration at baseline and at the end of intervention. C, PSQI scores at baseline and at the end of intervention. Note: PSQI, the Pittsburgh Sleep Quality Index. Intervention group is designated with a white square; control group is designated with a black square. *p<0.05. **p<0.01.

quality. To our knowledge, no published report has shown the effect of restricting nocturnal mobile phone use on pre-sleep arousal. Pre-sleep arousal was common among individuals with sleep disorders, and it was an important etiological factor of insomnia disorders [21]. It has been reported that insomnia patients have somatic, cognitive, and cortical arousal [22–25].

Sleep plays a very important role in mediating multiple domains of affective functioning including emotion, emotion memory, and emotion regulation. Previous studies have found the impact of sleep deprivation and sleep restriction on mood [26–29], and the influence of sleep extension on the reduction of the tendency to fall into a depressed mood [30]. Furthermore, all participants in the research agreed that using their mobile phones during bedtime affected sleep, and they had a high level of motivation to reduce use. When their goals are achieved, positive affect could increase and negative affect could decrease.

The participants of the intervention group had higher accuracy in the 1- and 2-back tasks and faster reaction times in the 0- and 1-back tasks. These results suggest that restricting mobile phone usage around bedtime could improve working memory. Previous studies on children and adolescents have demonstrated that sleep restriction and deprivation could affect

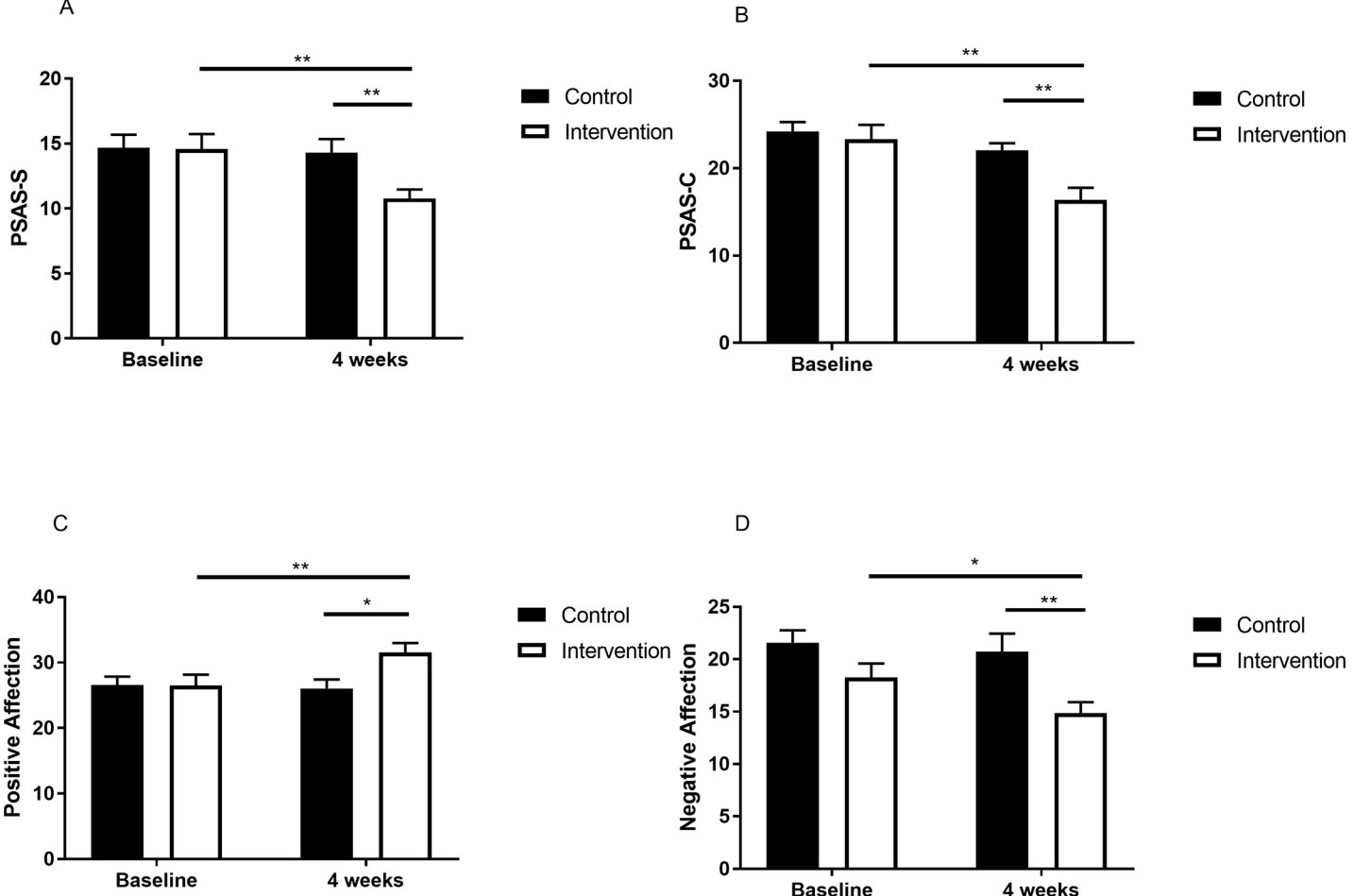

**Fig 3. The change in pre-sleep hyperarousal and mood after the intervention.** A, PSAS-S at baseline and at the end of intervention. B, PSAS-C at baseline and at the end of intervention. C, positive affection at baseline and at the end of intervention. D, negative affection at baseline and at the end of intervention. Note: PSAS-S, Pre-sleep Arousal Scale Somatic Arousal; PSAS-C, Pre-sleep Arousal Scale Cognitive Arousal. Intervention group is designated with a white square; control group is designated with a black square. *p<0.05. **p<0.01.

cognitive functions [31]. Increasing sleep duration could enhance cognitive performance [32]. In our study, sleep quality of individuals in the intervention group was shown to have improved significantly, having a positive effect on cognitive function.

Several limitations of the present study must be addressed. First, the present study used a sleep diary and PSQI to measure the participants' sleep, which may not be very objective. Further studies using objective measures such as actigraphy are needed for a more objective evaluation of sleep. Second, the participants of both groups agreed that excessive mobile phone use around bedtime could disturb their sleep and were all willing to reduce the amount of use around bedtime. However, these results might not be generally applicable to participants who did not agree that phone use around bedtime affects sleep and who lacked motivation to restrict mobile phone use around bedtime. Third, the sample size was small, thus limiting the interpretation of the results. These findings might set the basis for future studies.

In summary, restricting mobile phone use during bedtime for four weeks reduced sleep latency, pre-sleep arousal, and negative affect, and increased sleep duration, positive affect, and working memory. This intervention was effective in improving sleep quality and could be

A

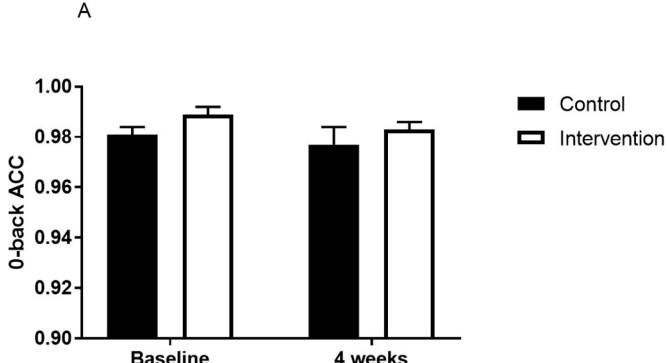

B

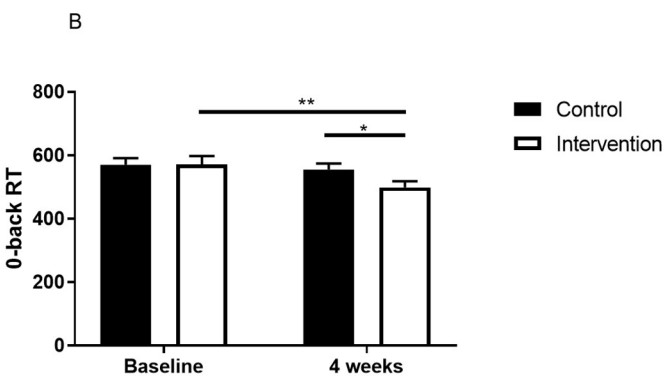

C

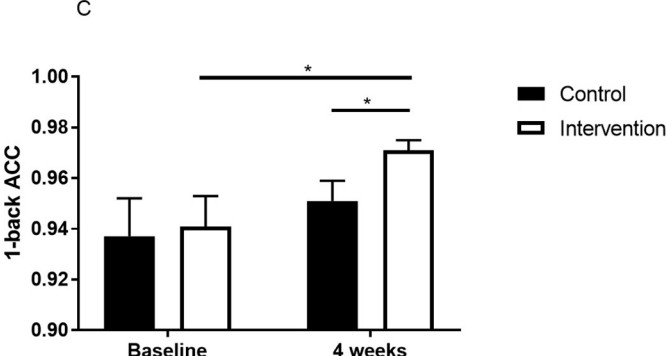

D

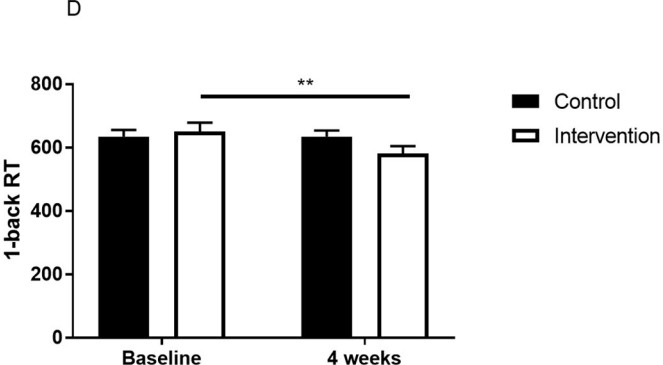

E

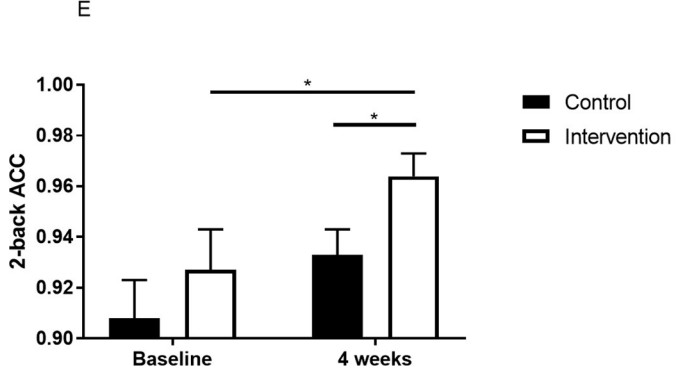

F

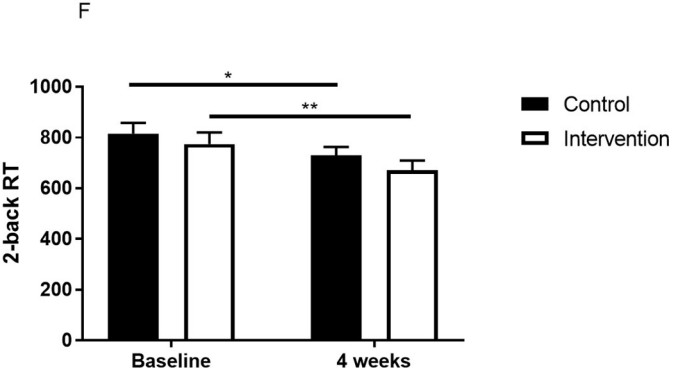

**Fig 4. The change in n-back after the intervention.** A, 0-back ACC at baseline and at the end of intervention. B, 0-back RT at baseline and at the end of intervention. C, 1-back ACC at baseline and at the end of intervention. D, 1-back RT at baseline and at the end of intervention. E, 2-back ACC at baseline and at the end of intervention. F, 2-back RT at baseline and at the end of intervention. ACC, accuracy; RT, reaction time. Intervention group is designated with a white square; control group is designated with a black square. *p<0.05. **p<0.01.

used as an adjunctive treatment for individuals with poor sleep quality and who are highly motivated to reduce bedtime mobile phone use.

## Supporting information

**S1 File. Data sets of the study.**
(SAV)

**S2 File. Questionnaire.**
(DOCX)

## Acknowledgments

We sincerely thank the students for their participation and all the medical staff involved in the testing. We would like to thank Editage (www.editage.cn) for English language editing.

## Author Contributions

**Conceptualization:** Jing-wen He.

**Data curation:** Jing-wen He, Lei Xiao.

**Formal analysis:** Jing-wen He, Zhi-hao Tu.

**Methodology:** Zhi-hao Tu, Tong Su.

**Resources:** Lei Xiao.

**Software:** Jing-wen He, Zhi-hao Tu.

**Supervision:** Yun-xiang Tang.

**Validation:** Yun-xiang Tang.

**Visualization:** Jing-wen He.

**Writing – original draft:** Jing-wen He.

**Writing – review & editing:** Lei Xiao, Tong Su, Yun-xiang Tang.

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
