## [Decision Letter · Decision Letter 0]

11 Oct 2019

PONE-D-19-19506

Effect of Restricting Bedtime Mobile Phone Use on Sleep, Arousal, Mood and Working memory

PLOS ONE

Dear Dr Tang,

Thank you for submitting your manuscript to PLOS ONE. After careful consideration, we feel that it has merit but does not fully meet PLOS ONE’s publication criteria as it currently stands. Therefore, we invite you to submit a revised version of the manuscript that addresses the points raised during the review process.

We would appreciate receiving your revised manuscript by Nov 25 2019 11:59PM. To enhance the reproducibility of your results, we recommend that if applicable you deposit your laboratory protocols in protocols.io, where a protocol can be assigned its own identifier (DOI) such that it can be cited independently in the future. For instructions see: http://journals.plos.org/plosone/s/submission-guidelines#loc-laboratory-protocols

We look forward to receiving your revised manuscript.

Kind regards,

Sergio Garbarino

Academic Editor

PLOS ONE

**Journal Requirements:**

2. Please note that there are still two instances (line 30, line 138) where 'diary' has been spelt 'dairy'. Please ensure these errors are corrected as part of your revisions.

3. Thank you for stating that “The funders had no role in study design, data collection and analysis, decision to publish, or preparation of the manuscript” in your financial disclosure.

Please also provide the name of the funders of this study (as well as grant numbers if available) in your financial disclosure statement.

**Comments to the Author**

1. Is the manuscript technically sound, and do the data support the conclusions?

Reviewer #1: Yes

Reviewer #2: No

2. Has the statistical analysis been performed appropriately and rigorously? 

Reviewer #1: I Don't Know

Reviewer #2: No

3. Have the authors made all data underlying the findings in their manuscript fully available?

Reviewer #1: Yes

Reviewer #2: Yes

4. Is the manuscript presented in an intelligible fashion and written in standard English?

Reviewer #1: No

Reviewer #2: Yes

5. Review Comments to the Author

Reviewer #1: In this study, He et al reported how reduction use of mobile phone at bedtime affected sleep through Pittsburgh Sleep Quality Index (PSQI), Pre-sleep Arousal Scale (PSAS), Positive and Negative Affect Schedule (PANAS) and n-back task, and concluded that reduction use of mobile phone at bedtime was very helpful to sleep, although the number of participant was not big enough (n=19).

Majors:

It is confused that several significance marks (* or **) were missing in Table 2 and Figs 2 and 3 according to results (Line 202-246). The writing is poor and tables are ugly.

Minors:

I suggest authors make all the statistical raw data available in spreadsheet format (like Microsoft excel) and describe statistical analysis in text with more details.

Figure legends are missing (only abbreviations).

Line 8 and 10: 1 and 2 should be superscript.

Line 30, 138: “dairy”?

Line 30-34: rephrase with ‘respectively’.

Line 82-83: rephrase

Line 94: screen should be screened

Line 94-99: confused, need rephrase

Line 135: missing original reference of PSQI

Line 137-138: Sleep diary data of the four weeks between that were collected or not?

Line 142: rephrase

Line 158: reference should be after period.

Line 161-162: “There are 10 questions for positive and 10 for negative affects”, redundant.

Line 172-173: “when the number presented is not identical to the previous one” replaced with “if not”.

Line 176-179: confused. Need rephrase.

Table 1 and 2: ugly, need revise.

Line 202-246 of results: Significance for some tests cannot be found in Table 2, need correct for each. For example:

Table 2: A star (*) is missing in the group effect of sleep latency and time effect of sleep duration according to Line 205 and 207-208, respectively.

Fig 2 and 3: Significance (* and **) should be noted with corresponding p-value for each test.

Line 212-219: correct Table 2 and Figs for significance

Line 222-225: correct positive affect for significance in Table 2 and Figs.

Line 227: it is not significant for p=0.06. Correct Table 2 and Figs.

Line 233-234, Line 236-237: Don’t know it is for accuracy or reaction time, or for 2-back accuracy?

Line 235: redundant with Line 232

Line 238-246: not corresponding to Table 2.

Line 262-264: “In this” study, “the effects” …. Was investigated.

Line 268: Sleep quality “was” improved significantly.

Line 269: One research “group”

Line 270-271: adolescents “who” stopped, delete “and”

Line 276-277: arousal “was” decreased, suggesting that it…

Line 278: no published report has shown

Line 281: It was reported that insomnia patients have somatic, cognitive and cortical arousal [21-24].

Line 282-283: “Content in the mobile….sleep quality” should be in Line 278.

Line 284-285: redundant. Sleep “plays”

Line 288-289: rephrase

Line 294: suggesting

Line 296-300: rephrase

Line 304-308: confused

Line 309: remove “four weeks of”, add “for four weeks” after “bedtime”.

Others:

1. The study presents the results of original research.

Yes.

2. Results reported have not been published elsewhere.

Yes.

3. Experiments, statistics, and other analyses are performed to a high technical standard and are described in sufficient detail.

Ok.

4. Conclusions are presented in an appropriate fashion and are supported by the data.

Ok.

5. The article is presented in an intelligible fashion and is written in standard English.

Ugly tables. Poor writing.

6. The research meets all applicable standards for the ethics of experimentation and research integrity.

Yes.

7. The article adheres to appropriate reporting guidelines and community standards for data availability.

Yes.

Reviewer #2: 1) Authors should briefly provide a description/synonym of netizens, since not all readers may be familiar with this term.

2) There are some typos. Some sentences begin with "And". English should be revised for the sake of clarity.

3) Only after reading methods, I come to know that the study is a randomized trial. However, authors did not follow reporting guidelines for randomized trials (for example, the title does not contain informative keywords). Did authors register the randomized trial in a database of RTs?

4) Sample size is very small (because was small already the recruited sample size at the commencement of the trial, after a lot of individuals being excluded). However, authors performed a lot of statistical analyses. Seems (to my opinion) that the study is underpowered to make sufficiently strong statements and conclusions. Is it a pilot study? How much is the statistical power found (in tems of effect sizes)? And the post-hoc power Did authors perform a sample power analysis?

5) Some important studies are not mentioned (one for all: Mobile Phone Use and Mental Health. A Review of the Research That Takes a Psychological Perspective on Exposure. Thomée S. Int J Environ Res Public Health. 2018 Nov 29;15(12)).

6) Graphs miss bars of standard deviations.

6. PLOS authors have the option to publish the peer review history of their article (what does this mean?). If published, this will include your full peer review and any attached files.

Reviewer #1: Yes: Guo Luo

Reviewer #2: No

---

## [Author Response · Author response to Decision Letter 0]

8 Nov 2019

Thank you for your comments concerning our manuscript entitled “Effect of Restricting Bedtime Mobile Phone Use on Sleep, Arousal, Mood and Working memory” (ID: PONE-D-19-19506). Those comments are valuable and very helpful for revising and improving our paper, as well as the important guiding significance to our researches. We have modified our manuscript according to the comments. Besides, we have carefully checked through the whole manuscript and corrected some grammar mistakes. We hope the revised paper would meet the requirement from you.

---

## [Decision Letter · Decision Letter 1]

23 Jan 2020

Effect of restricting bedtime mobile phone use on sleep, arousal, mood, and working memory: A randomized pilot trial

PONE-D-19-19506R1

Dear Dr. Tang,

We are pleased to inform you that your manuscript has been judged scientifically suitable for publication and will be formally accepted for publication once it complies with all outstanding technical requirements.

With kind regards,

Sergio Garbarino

Academic Editor

PLOS ONE

Additional Editor Comments (optional):

Reviewers' comments:

Reviewer's Responses to Questions

**Comments to the Author**

1. If the authors have adequately addressed your comments raised in a previous round of review and you feel that this manuscript is now acceptable for publication, you may indicate that here to bypass the “Comments to the Author” section, enter your conflict of interest statement in the “Confidential to Editor” section, and submit your "Accept" recommendation.

Reviewer #1: All comments have been addressed

Reviewer #2: All comments have been addressed

2. Is the manuscript technically sound, and do the data support the conclusions?

Reviewer #1: (No Response)

Reviewer #2: Yes

3. Has the statistical analysis been performed appropriately and rigorously? 

Reviewer #1: (No Response)

Reviewer #2: Yes

4. Have the authors made all data underlying the findings in their manuscript fully available?

Reviewer #1: (No Response)

Reviewer #2: Yes

5. Is the manuscript presented in an intelligible fashion and written in standard English?

Reviewer #1: (No Response)

Reviewer #2: Yes

6. Review Comments to the Author

Reviewer #1: The authors should carefully check grammar/typos again before print.

Reviewer #2: All comments and concerns of reviewers have been properly addressed and the manuscript has been considerably improved.

7. PLOS authors have the option to publish the peer review history of their article (what does this mean?). If published, this will include your full peer review and any attached files.

Reviewer #1: No

Reviewer #2: Yes: Nicola Luigi Bragazzi

---

## [Editor Report · Acceptance letter]

3 Feb 2020

PONE-D-19-19506R1 

Effect of restricting bedtime mobile phone use on sleep, arousal, mood, and working memory: A randomized pilot trial 

Dear Dr. Tang:

I am pleased to inform you that your manuscript has been deemed suitable for publication in PLOS ONE. Congratulations! Your manuscript is now with our production department. 

With kind regards,

on behalf of

Dr. Sergio Garbarino 

Academic Editor

PLOS ONE